# Review of Clinically Assessed Molecular Fluorophores for Intraoperative Image Guided Surgery

**DOI:** 10.3390/molecules29245964

**Published:** 2024-12-18

**Authors:** Yuan Ge, Donal F. O’Shea

**Affiliations:** Department of Chemistry, RCSI, University of Medicine and Health Sciences, 123 St Stephen’s Green, Dublin 2, D02 YN77 Dublin, Ireland

**Keywords:** molecular dyes, near-infrared fluorescence, fluorophores, clinical trials, surgery

## Abstract

The term “fluorescence” was first proposed nearly two centuries ago, yet its application in clinical medicine has a relatively brief history coming to the fore in the past decade. Nowadays, as fluorescence is gradually expanding into more medical applications, fluorescence image-guided surgery has become the new arena for this technology. It allows surgical teams to real-time visualize target tissues or anatomies intraoperatively to increase the precision of resection or preserve vital structures during open or laparoscopic surgeries. In this review, we introduce the concept of near-infrared fluorescence guided surgery, discuss the recent and ongoing clinical trials of molecular fluorophores (indocyanine green, 5-aminolevulinic acid, methylene blue, IR-dye 800CW, pafolacianine) and their surgical goals, highlight key chemical and medical factors for imaging agent optimization, deliberate challenges and potential advantages, and propose a framework for integrating this technology into routine surgical care in the near future. The notable clinical achievements of these fluorophores over the past decade strongly indicates that the future of fluorescence in surgery is bright with many more patient benefits to come.

## 1. Introduction

Fluorescence is the light emission by an atom or molecule due to electronic transition, which is stimulated by the absorption of electromagnetic energy from external radiation. This phenomenon was first formally reported in 1565, but the term “fluorescence” was first proposed by George Gabriel Stokes in his seminal paper entitled “On the Change of Refrangibility of Light” published in 1852 [1]. From that time, fluorescence has been applied as an investigative tool in many different areas, from materials technology to life sciences. In 1953, fluorescent dyes were first applied in medicine, using berberine sulfate, acid fuchsin, and acridine to identify cancer cells in vaginal smears by staining their nucleic acids [2]. Over the following half century, fluorescence gradually expanded into more medical studies, including investigations for tumor indication, cardiac ischemia evaluation, bile duct visualization, and bowel perfusion assessment. Nowadays, fluorescence image-guided surgery has become the new arena for this technology. It offers the generalizable potential of surgeons to real-time visualize the target tissue intraoperatively in order to increase the precision and safety of surgical tissue resection and can be performed during either open or laparoscopic surgeries.

Generally, the wavelengths associated with molecular absorption and fluorescence range from 200 to 1000 nm. The earlier fluorophores investigated had mostly mid-range emission wavelengths, for example, the antibiotic drug tetracycline with λ_max_ of 520 nm and natural and synthetic porphyrins giving a red fluorescence reading in the 600–650 nm range when excited by ultraviolet light (400 nm). These early fluorophores often suffered from disadvantages such as poor water solubility, poor photophysical stability, low extinction coefficient and fluorescence quantum yield, and high toxicity precluding their translation into clinical use [3]. Therefore, a fluorophore with clinical potential should not only overcome these drawbacks but also demark the tissue of interest with excellent fidelity and have an observation window that fits within the time frame of the operation. In the last decade, studies on this area have concentrated on near-infrared (NIR) fluorophores emitting photons within the 700–900 nm range, which is known as the in vivo optical window. In comparison to fluorescence in the visible region, NIR fluorescence has the advantages of enhanced tissue penetration up to centimeters, less light scattering, and avoidance of tissue auto-fluorescence providing higher signal-to-noise ratio and imaging resolution.

To date, fluorophores have been specifically designed for many different tissues for disease treatment, including tumor detection, sentinel lymph nodes mapping, neurological diseases, cardiovascular diseases, bile duct identification, perfusion assessment, and gland visualization. To date, the US Food and Drug Administration (FDA) has approved four fluorophores for clinical use, two of which are historical approvals, indocyanine green (ICG), and methylene blue (MB), in addition to 5-aminolevulinic acid (ALA) which is a biosynthetic precursor to propotoporphyrin IX (PpIX) and pafolacianine (OTL38) [4,5]. While the hardware and software for intraoperative image capture and display has rapidly advanced in recent times, a major remaining limitation for fluorescence guided surgery is the availability of clinically approved dyes. This has encouraged research from various groups dedicated to designing the chemical structure and optimizing the optical properties of fluorophores to meet the specific requirements for imaging agents used in surgery. In this review, we focus on molecular fluorophores that have entered clinical trials or been used with clinical approval for surgeries in the last decade, list their advantages and limitations for intraoperative use, and outline the key points to their optimization as imaging agents, in order to issue a framework to make this technology more mature for routine care of patients in future medicine. By providing a comprehensive analysis of these fluorophores, this review aims to guide researchers and clinicians in selecting and refining imaging agents, ultimately improving surgical precision and patient outcomes.

The content used in the review was sourced from the academic and patent literature using the search strategy outlined below:Literature searches were conducted across multiple academic and medical databases to ensure comprehensive coverage, including SciFinder, PubMed, Web of Science, Google Scholar, Clinical Trials Gov.Searches employed different combinations of keywords such as: fluorescence guided surgery, near-infrared fluorescence, fluorophores, ICG, 5-ALA, methylene blue, IR-Dye 800CW, pafolacianine, clinical trials in fluorescence imaging.Criteria for inclusion and exclusion included peer-reviewed publications or clinical trials focused on direct applications in surgery from the last 10 years. Publications lacking detailed clinical data were not included.

## 2. Indocyanine Green (ICG)

Cyanines are named for a systematic family of dyes with structures containing two heterocyclic moieties linked by polymethine chains of varying length. The use of cyanine dyes in industry dates back over a century, starting with their application as sensitizing dyes in photographic silver halide film and later evolving to roles in fluorescent microscopy, biomedical labeling, and assaying.

In the example shown in Figure 1, when *n* = 0, the dye is yellow and provides sensitization to blue light. When *n* = 1, the dye, a carbocyanine, is magenta and absorbs green light. For *n* = 2, a dicarbocyanine, the dye appears cyan and sensitizes silver halide to red light [3]. Each iterative step of extending the double bond conjugation is reflected in an increased absorption wavelength maxima of the resulting chromophore. Nowadays, a systematic series of more than 20,000 cyanine dyes has been developed and depending on the length of the polymethine chain, their electromagnetic spectrum ranges from the NIR to UV.

Indocyanine green (ICG) is an amphiphilic water-soluble cyanine dye, with *n* = 3, having absorbance and emission maxima in the spectral range of 779 to 825 nm, with a molecular weight of 751.4 Da [6] (Figure 2). Recently, a revised streamlined synthesis of it was reported in which the last step was achieved in excellent yield (Figure 2) [7]. It was firstly developed by Heseltine and Brooker in the Kodak Research Laboratories in 1955 [8], being adapted from a similar dye being used for aerial photography [9]. From 1956, in collaboration with medical researchers in the Mayo Clinic, this “new dye” was already being explored for use in cardiac output measurement [10] and then extended to hepatic function studies in 1958 [11]. In 1959, ICG was approved by the FDA for determining cardiac output, hepatic function and liver blood flow, and for ophthalmic angiography [6,12].

To date, ICG remains the most commonly used clinical fluorophore as its emission spectral range provides a higher signal-to-background ratio due to lower tissue autofluorescence and increased light tissue penetration [13]. It has poor solubility in pure water but binds to the blood plasma albumin protein giving a strong emission [14] and is excreted, without chemical change, solely via the liver. Besides its preferable fluorescent characteristics, the widespread acceptance of ICG is also attributed to its confinement within the vascular compartment through binding to plasma proteins, fast and almost exclusive excretion into bile (in vivo half-life of 2.5–3.0 min), low toxicity, and safeness to be used with low incidence of allergic reactions [6]. This has given rise to a clinical acceptance of ICG’s potential for sentinel lymph node (SLN) mapping, surgery guiding by intraoperative cholangiography, and for perfusion assessment during abdominal and reconstructive surgeries.

### 2.1. ICG for Sentinel Lymph Node (SLN) Mapping

Since 2010, ICG fluorescence imaging for SLN detection or biopsy, have been applied in breast, gastric, melanoma, vulvar, cervical, endometrial, non-small cell lung, oropharyngeal, colorectal and prostate cancers (see Appendix A for the literature references). These trials have demonstrated ICG imaging to be a safe, efficient, and reliable method for SLN identification. Compared with the conventional colorimetric SLN mapping technology using non-fluorescent blue dyes or radioactive agents such as 99m Tc nanocolloid, ICG can provide a higher detection rate and increased sensitivity [15]. Intraoperatively this is achieved through an injection of ICG into the suspect tissue, with drainage into the localized lymphatics occurring within minutes, allowing real-time visualization of lymphatic channels and vessels, thereby informing the decision making process of the surgical team.

From 2012 to 2015, a multicentre cohort study was carried out in the United States with 385 patients enrolled to compare ICG-SLN mapping with the gold standard of complete lymphadenectomy in endometrial cancer. The results indicated that 293 of the patients had at least one SLN detected successfully using ICG-SLN mapping, 257 of whom were node negative and 36 were node positive, and 35 of 36 patients had metastatic disease correctly identified in the SLN. The conclusion of the trial stated that “a sensitivity to detect node-positive disease of 97.2% and a negative predictive value of 99.6%”, was achieved, showing that the ICG-SLN detection is an alternative option to lymphadenectomy in endometrial cancer [16].

Yet this does not mean that it offers improved outcomes for all cancers. As Klode et al. has reported, among 80 patients with malignant melanoma, the result of intraoperative visualization of SLN by ICG was 21% before incision and 96% after skin incision, so that the use of ICG for SLN detection was limited for melanoma patients compared with 100% detection using a technetium Tc 99m guided technique [17]. This outcome demonstrates that emission light scattering by tissue must be specifically considered for each clinical use. ICG’s lymphatic imaging property has also been used for postoperative lymph flow evaluation, lymphedema management, and to locate the lymphatic abnormalities to investigate the impact of gene mutations on the lymphatic system (see Appendix A for additional literature references). Chemical efforts to improve ICG results were reported by Vahrmeijer et al. in which they compared ICG pre-mixed with human serum albumin (HSA) versus ICG alone for LN detection. ICG-HSA solutions were prepared by pre-mixing aqueous solutions of ICG and HSA before localized tissue injections. Trial results using preformed HSA-ICG for SLN detection in cervical, vulvar, and breast cancer patients found no significant difference between it and using ICG alone, therefore concluding that administrating ICG-HSA had no advantage over ICG alone [18,19,20].

### 2.2. ICG for Surgical Decision Guidance

#### 2.2.1. ICG for Bile Duct Identification

ICG imaging has been assessed during laparoscopic cholecystectomy (LC) and pancreaticoduodenectomy (see Appendix A for additional literature references) operations to delineate the bile ducts to reduce the risk of accidental injury such that the associated patient complications could be prevented. Schols et al. has conducted a feasibility study of ICG imaging during LCs, and in 2013 they presented their results of a 30-patient study showing that 83% of common bile ducts and 97% of cystic ducts were clearly identified at significantly earlier stages of the operation than the conventional camera mode [21]. Based on the conclusion of this feasibility study, a multicentre randomized controlled clinical trial was carried out to assess ICG imaging as a means of reducing vascular and bile duct injuries. The results published in 2023 reflect that, with 294 patients involved, ICG fluorescence imaging in LC was beneficial for an earlier identification of the hepatic biliary anatomy [22].

A retrospective review has collected the data of 1389 patients who underwent laparoscopic cholecystectomy with or without ICG imaging in a single academic center from 2013 to 2019. Upon analyzing all the statistics, it was determined that ICG cholangiography significantly improved patient outcomes by reducing operational times, decreasing conversion to open procedures, and shortening hospitalization durations [23]. Several additional clinical trials have proposed that ICG imaging could hold increased promise for difficult or emergency LC, such as acute cholecystitis (see Appendix A for additional literature references).

#### 2.2.2. ICG for Intraoperative Tumor “Light Up” Detection

Even though ICG has had FDA approval for many decades, its non-tissue specific accumulation (other than the liver) makes it unsuitable for intraoperative tumor detection.

As ICG rapidly saturates the liver upon administration, remarkably it has been discovered that following its excretion small amounts are retained in normal tissue surrounding metastatic sites embedded within the liver. This opened the clinical opportunity for its use to identify and guide the surgical removal of these distributed cancerous lesions. In this way, ICG intraoperative imaging has also successfully detected lung and liver metastases lesions (see Appendix A for additional literature references), some of which were even missed or undetected by other modalities, with the appearance of clear fluorescent rims around the tumors (Figure 3) [24]. In clinical practice, the time between ICG intravenous administration and optimal imaging has been investigated with varying periods identified as 24 h or 48 h prior to surgery [24], but others reported that guiding ICG fluorescence is still usable at 6 [25] and 14 days [26] post-administration. The mechanism of why ICG can be retained for such a long time is still debated. In one study, the ICG accumulation in the transition area between tumors and normal liver tissue was explored with the conclusion reached that “the pattern of rim fluorescence could be explained by the presence of immature hepatocytes in the liver tissue surrounding the tumor that have taken up ICG, but exhibit impaired biliary clearance” [24]. Though ICG imaging can detect overlooked lesions by other methods, there is also a relatively high false-positive proportion which cannot be ignored [26,27].

This concept of dye retention around metastatic lesions has also been investigated for lung cancer patients; ICG was able to intraoperatively “light up” normal parenchyma in lung with sufficient fluorescent density, but left the metastatic lesions far less fluorescent [28]. But this failed for pancreatic cancers as no distinct contrast between the tumor and pancreas was observed [29] and was deemed not suitable for ovarian cancer metastases with a high number of false-positives recorded [30].

#### 2.2.3. ICG for Tissue Perfusion Assessment

The perfusion of blood through tissue directly relates to the health status of that tissue so an assessment of viability can be obtained through the fluorescence visualization of ICG perfusion in the vasculature of a tissue of interest. By the function of evaluating microcirculation and tissue viability, ICG has been used to qualify tissue perfusion in reconstructive surgeries and tissue transplantations (see Appendix A for the literature references). It enabled a risk prediction of post-operative tissue necrosis, thereby achieving a reduction in patient complications being avoided due to the interoperative identification of inadequate perfusion.

In a multicentre phase II trial of ICG imaging during colorectal surgery with 504 patients (330 neoplastic cancers and 174 benign cancers) involved, ICG was administrated at two critical intraoperative time points. Firstly, ICG was injected before bowel resection and images were acquired within one minute to assess the colonic tissue perfusion (Figure 4b), and the surgeons would adjust the transection point referring to whether the perfusion signal reached the planned transection site. After the construction of the anastomosis (reattachment of the two ends of the intestine), ICG was injected again for the assessment of the vascular sufficiency in this region of connected tissue (Figure 4c), and anastomosis would be redone if the perfusion proved to be insufficient or a leak identified. With the assistance of ICG imaging, the anastomotic leak rate was decreased from 5.8% to 2.6%, and no anastomotic leaks were found for the 29 patients whose surgical plans were revised based on ICG intraoperative perfusion assessment [31]. This positive result was also revealed in a multi-institutional study conducted in 2014, in which ICG fluorescence was used to assess colon and rectal perfusion during colorectal resection. Eleven out of 147 patients had the transection site altered and no anastomotic leaks occurring [32]. A recent trial showed good potential for the use of an artificial intelligence-(AI) based analysis method of ICG angiography to predict anastomotic complications after laparoscopic colonic surgery. While additional clinical trials are necessary, the authors conclude that the analysis of perfusion during surgery may reduce the probability of post-laparoscopic colorectal anastomotic complication [33].

With a randomized double-blinded controlled trial among 95 patients undergoing abdominal wall reconstruction (AWR), ICG was intravenously injected and images were used to evaluate perfusion levels with any deficits created by hernia or previous incisions identified. In an experimental group, surgeons planned their incision according to the perfusion deficits displayed on the ICG imaging, meanwhile in a controlled group, surgeons made their incision plan depending solely on their clinical judgment. The results revealed that ICG fluorescence imaging did not reduce wound-related complications or reoperations, but it did predict the risk of wound infection by successfully identifying areas of poor tissue perfusion [34].

In a first of its kind trial, Cahill et al. demonstrated the successful application of AI interpretation of the ICG perfusion fluorescence profiles as ICG first passes through the suspect area containing both normal and cancerous tissues. As physical and biological properties of cancerous and normal tissue differ, it was anticipated that this could be detectable through the AI analysis of the dynamic emission patterns as ICG first passed through the tissue. In the trial following ICG intravenous injection, the dynamic changes in ICG emission profile over 90 s was recorded in sections of colon tissue suspected to have cancerous regions. Using these data, biophysical models of an AI classifier could identify differences between normal and cancerous tissue regions, not detectable by human vision (Figure 5). The speed of fluorescence data acquisition and AI analysis allowed tissue classifications to be produced intraoperatively without impeding the normal surgical workflow. The outcome of the trial was that 19 out of 20 cancers were correctly diagnosed, with a high accuracy of 95% [35]. This technology could empower surgeons with data-driven tissue classifications during the operation, potentially improving surgical decisions. This marks a significant advancement over traditional imaging approaches as the tissue information is obtained without needing human observable differences in fluorescence intensity patterns at a single time point but rather through AI-assisted understanding of dynamic tissue perfusion [36,37]. It was proposed that this approach could be applicable for use with other fluorophores as well as other tissue types.

## 3. 5-Aminolevulinic Acid (5-ALA)

5-Aminolevulinic acid (5-ALA) is a colorless metabolic amino acid intermediate along the biosynthetic pathway to protoporphyrin IX (PpIX), which is fluorescent (Figure 6). The heme bio-synthetic pathway of which PpIX is an advanced intermediate begins with glycine and succinyl-CoA with 5-ALA being the product from their combination [38]. A further seven biosynthetic stages are required to convert 5-ALA into PpIX with some key intermediates shown in Figure 6. This synthesis of 5-ALA is catalyzed by ALA synthase and is the rate-limiting enzyme of the pathway and under the negative feedback control of heme. By exogenously introducing 5-ALA, unnaturally elevated levels of PpIX occur as a deficiency of iron supply limits heme synthesis giving rise to PpIX accumulation within cells.

In clinical use, patients are orally administered 5-ALA (20 mg/Kg) as an aqueous solution typically two to four hours prior to surgery. This introduction of 5-ALA stimulates the over-production of PpIX within some cancer cells, allowing for them to be distinguished from normal cells through the fluorescence signature of PpIX. The emission of PpIX in aqueous solutions and tissues ranges from 620 to 635 nm [39] which is notably shorter by approximately 190 nm than ICG. This significantly restricts its use to tissues which are less pigmented such as brain, bladder, or skin where the competing autofluorescence of heme is less pronounced.

In early works, several research groups demonstrated that 5-ALA conversion to PpIX produced high fluorescence in experimental in vivo models of malignant gliomas, the most common brain tumor in adults. In anticipation that this could help distinguish the rapidly proliferating tumor boundary intraoperatively, Stummer et al. trialed 5-ALA for intraoperative detection of malignant gliomas in a series of clinical studies [40,41] (see Appendix A for additional literature references). In 2006, a randomized controlled multicentre phase III trial using PpIX guided surgery was reported on for resection of high-grade malignant gliomas with the clinical data collected from 270 patients. It was demonstrated that 5-ALA induced fluorescence guided surgery can be beneficial for patients in terms of completeness of tumor removal and progression-free survival [42].

Since then, 5-ALA has been widely applied to the resection of brain and spinal tumors, such as gliomas, glioblastoma, meningioma, and brain metastases, but with varying success as it was found to perform differently among various tumors (see Appendix A for additional literature references). For gliomas, researchers found that only fast growing, high-grade gliomas produced sufficient amounts of PpIX to be microscopically visualized [43], whereas for others, the fluorescence emission intensity remained below the detection threshold [44,45]. Although, it has been proposed that 5-ALA in combination with intraoperative confocal microscopy can improve its detection [46]. For meningiomas, three studies showed fluorescence appeared, respectively, 87.5%, 92.6%, 89.9% in all meningiomas, even with the low-grade cases [47,48,49]. Also, in a small trial of 12 patients with low grade bone invasion meningiomas, PpIX fluorescence was detected in all tumors [50]. Yet, in cases of cerebral metastases, originating from various primary cancers, did not consistently exhibit fluorescence [51]. In 2017, the FDA approved 5-ALA as “an optical imaging agent indicated in patients with high grade gliomas as an adjunct for the visualization of malignant tissue during surgery” [52].

The reason for success or failure of PpIX differentiation of different grade brain tumors remains unproven; researchers believe the blood–brain barrier breakdown is a key factor [53], but other factors such as ferrochelatase deficiency or other molecular pathway changes may also be contributing [54,55,56,57]. Some of the literature reports showed that the absence of fluorescence did not correlate with the absence of tumor, and conversely, some fluorescent tissues were false positives, i.e., non-tumorous upon histological assessment [58]. A prospective phase II clinical trial was raised to assess the correlation of intraoperative fluorescence intensity with the pathology assessment of the resected tissue for high-grade glioma surgeries [59]. From the results of 211 biopsies from 59 patients, high fluorescence intensity, as intraoperatively evaluated by surgeons, correlated with 97.4% with tumor cell abundant tissue, as evaluated by pathologists. But the absence of fluorescence correlated with the absence of tumor for only 37.7% of the tissues examined and 35.4% non-tumor biopsies were fluorescent [59]. While the use of 5-ALA has undoubtedly being a clinical success there remains room for future improvements perhaps from both the fluorophore itself and the image capture systems used.

Though 5-ALA is most commonly used for brain tumor resections, it has also been trialed for breast and bladder cancers (see Appendix A for the literature references). It was found that with a dosage of 15 mg/kg 5-ALA, breast tumor tissue exhibited red fluorescence 2.5 to 5 h post administration and the positive predictive value for detecting breast cancer inside and outside the grossly demarcated tumor border was 100% and 55.6%, respectively [60].

## 4. Methylene Blue (MB)

The chemistry of methylene blue (MB) has a long history, first prepared in 1876 by Heinrich Caro, it has been referred to as the first fully synthetic drug used in medicine (Figure 7). It is on the WHO list of essential medicines and over the years it has been used for the treatment of malaria, urinary tract infections, and cyanide poisoning though currently it is primarily recommended to treat methemoglobinemia.

As a strongly blue colored fluorescent dye, MB has been applied to clinical imaging since being introduced by Keaveny in 1968 for identification of parathyroid adenoma [61] as it can be up taken by endocrine tissues easily. From then on, MB was widely used to aid naked eye identification of blue colored parathyroid glands or insulinomas during thyroid or pancreatic surgeries, but this required a high-dose intravenous injection (3–7.5 mg/kg) [62,63]. High-dose MB not only stains the operative field, but also brings a significant risk of severe adverse events, including toxic metabolic encephalopathy [64] and exerting neurotoxic effects on the central nervous system [65].

MB is also a moderately strong fluorophore emitting in the 650 to 750 nm range, even at low doses, with researchers recently investigating its potential clinical uses. Trials from researchers in Leiden University studied the possibility of using low-dose MB (0.5 mg/kg) to identify parathyroid adenomas intraoperatively [66]. Ten out of twelve patients were histologically confirmed with parathyroid adenoma, and nine of these ten were identified by MB fluorescence. This was a significant improvement on the 70% preoperative detection result using a 99mTc-sestamibi single photon emission CT scan [66]. From 2014, a phase 1b, interventional study was carried out in the UK involving 41 patients in order to develop a protocol for thyroid and parathyroid imaging surgery with the outcome identifying 0.4 mg/kg as the optimum dose [67,68].

As MB is excreted via the kidneys, this endues it with the additional use of intraoperative identification of ureters, the smooth muscle tubes that transport urine from the kidneys to the bladder. Accidental ureteric injury during abdominal surgery is a serious complication so it would be of beneficial surgical assistance if patient ureters could be fluorescence identified intraoperatively (see Appendix A for additional literature references). Barnes et al. reported 64 out of the 69 ureters were successfully identified by MB fluorescence, of which, 14 were undetectable under white light. Additionally, out of the 50 ureters observed by both MB fluorescence and white light, 14 were seen earlier with fluorescence, and the optimum intravenous dose was 0.75 mg/kg [69].

MB has also been investigated clinically for neuroendocrine and breast tumor imaging though more trials would be needed to determine if patient benefit is achievable (see Appendix A for additional literature references).

## 5. IR-Dye 800CW

While ICG is the most widely clinically used fluorophore, its short half-life and fast excretion rate via the hepatobiliary system severely limits its broader use beyond existing applications. Therefore, researchers have designed new bio-conjugated (antibodies, peptides, etc.) fluorophores to enhance their affinity for specific cancer types. IR-Dye 800CW has similar excitation and emission wavelength ranges (780–900 nm) as ICG, but unlike ICG which has no reactive functional group, it can be substituted through conversion of the carboxylate to an activated ester and coupling with amines (Figure 8) [70].

In the last decade, clinical trials have reported IR-Dye 800CW labeled to the antibodies cetuximab or panitumumab as a means of targeting EGFR (epidermal growth factor receptor) expressing tumors and bevacizumab for VEGF-A for vascular endothelial growth factor. These constructs have been widely clinically explored for head and neck cancers or high-grade glioblastoma, breast, esophagus, rectal/colon, and pancreatic cancers (see Appendix A for additional literature references). These conjugates can be used to bias their retention within cancers for metastatic disease identification intraoperatively. But as it can take a prolonged time to achieve sufficiently high tumor to background fluorescence ratios, surgeries may have to be scheduled 1 to 7 days after the intravenous infusion of the conjugate. The uncertainty of knowing when an operation should proceed such that it occurs at the optimal tumor-to-background fluorescence ratio could make the broad surgical acceptance of this approach a challenge.

## 6. Pafolacianine (OTL38)

In late 2021 and 2022, the FDA approved the NIR fluorescent dye pafolacianine for adult patients with ovarian and lung cancers as an imaging agent for intraoperative identification of malignant lesions [5,71]. Pafolacianine, also known as OTL-38, is a folic acid conjugate of the meso-chloro substituted heptamethine cyanine dye, S0456 (Figure 9). The conjugation reaction involved the meso-chloro displacement of S0456 by a nucleophilic phenoxy moiety to install the folic acid group through a phenoxyvinyl ether bridged cyanine [72]. With an emission maximum at 793 nm in DMSO, its spectral wavelengths are similar to those of ICG. It shows specificity and affinity for folate receptor α, which is expressed in over 90% of epithelial ovarian cancers and lung cancers [73,74]. Clinically it is administered intravenously one to nine hours prior to surgery, allowing surgeons to visualize tumor margins intraoperatively using standard NIR imaging systems [75].

The use of pafolacianine for ovarian cancer was studied in phase 2 [76] and 3 [77] trials. Results from the phase 2 stage showed it successfully identified 85% of tumors with a positive predictive value of 88%. Additionally, 48% of patients had at least one extra lesion detected exclusively through its use [78]. The phase 3 results demonstrated sensitivity to detect ovarian cancer at 83% and the patient false-positive rate was 24.8%. Encouragingly, 33% of patients had additional cancerous tissue detected outside the planned resection area, which were not identified through white light assessment and palpation [79].

The results of a phase 3 lung cancer trial [80], of which 100 patients with known or suspected lung cancer received a dose of OTL38 before they were evaluated, demonstrated favorable efficacy and safety in patients undergoing thoracic surgery. 19% of patients had additional cancerous lesions that were not observed by standard visual or tactile inspection [81]. Additional indications investigated include the detection of LNs and metastases of renal cell carcinoma, bladder cancer, and pituitary adenomas (see Appendix A for additional literature references).

## 7. Conclusions

Among the fluorophores discussed in this review, ICG [82] and methylene blue are the most cost-effective and widely accessible worldwide with 5-ALA resulting in only a moderate increase in hospital costs relative to conventional surgery using white light imaging [83]. As relatively new agents, IR-Dye 800CW and pafolacianine face accessibility challenges due to high costs or limited regulatory approval, which may hinder their routine clinical use on a global scale.

While the chemistry of NIR fluorophores has significantly advanced fluorescence-guided surgery, enhancing both precision and effectiveness, challenges still persist. Future development in NIR fluorophores is trending towards more targeted, precise, and versatile imaging solutions. Continuous refinement of these agents aims to address issues related to tissue specificity, contrast, and real-time imaging capabilities. An important recognized design feature being the need to eliminate prolonged delays between fluorophore administration and imaging, with the ideal scenario being that both administration and imaging occur during the operation without impeding the normal surgical workflow. It seems highly likely that the adoption of AI for the interpretation of dynamic images is set to change the way in which information is presented to the surgical team. Rather than relying on human interpretation of static images, AI analysis of dynamic changes in images could be employed to intraoperatively produce classifying maps of the tissue undergoing resection.

To produce a new clinically viable fluorophore requires the interdisciplinary collaboration of chemistry and medicine to work as closely as possible such that fluorophore design is fully aligned with the surgical application from the outset and can be tested preclinically and clinically to demonstrate its safety and efficacy. The entire process from preclinical testing to final FDA approval can take 7–10 years or more, depending on the complexity of the fluorophore, the surgical procedure and the safety data required.

While not a focus of this review, imaging hardware compatibility is a significant technological consideration for the clinical adoption of fluorescence-guided surgery, particularly across different institutions and hospitals. Different fluorescence imaging systems from different manufacturers do not necessarily have identical fluorophore excitation and emission collection parameters and each will use propriety software for raw data interpretation and image display. Establishing industry-wide standards for image data collection, processing and display is essential for the broader adoption and consistency in fluorescence-guided surgery.

As the dye chemistries, imaging hardware and AI models evolve, the combined integration of these advanced technologies into clinical practice is expected to improve surgical outcomes and patient safety, marking a significant leap forward in the field of surgical practice and beyond. We hope this review will encourage more researchers to focus on overcoming key translational barriers, such as false positives, tissue specificity, and administration delays, which are critical to realizing the full potential of these technologies in clinical use.

## Figures and Tables

**Figure 1 molecules-29-05964-f001:**
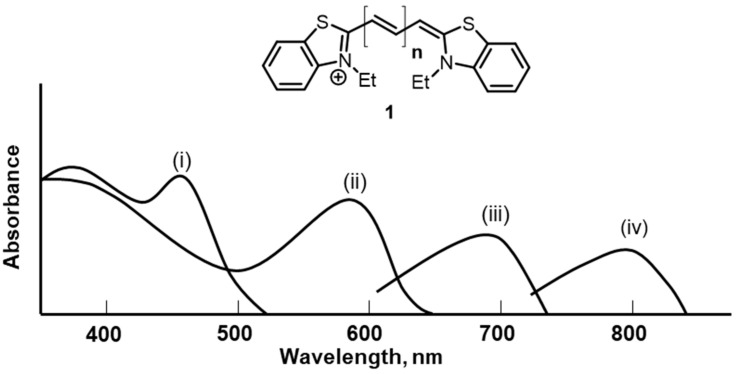
Representative cyanine dye structure **1**. Absorption curves for **1** when (i) corresponds to *n* = 0; (ii) to *n* = 1; (iii) to *n* = 2; and (iv) to *n* = 3 [3].

**Figure 2 molecules-29-05964-f002:**
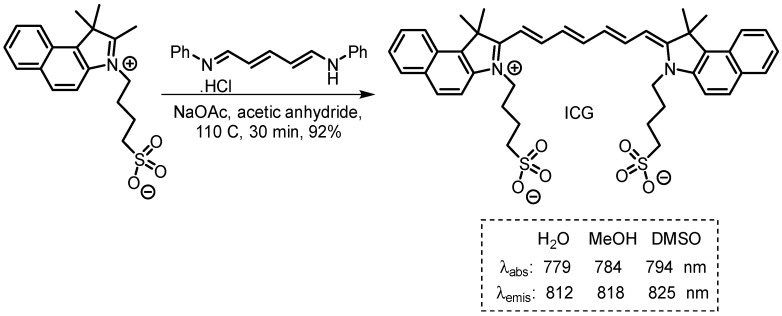
Synthesis, structure, and photophysical properties of indocyanine green (ICG).

**Figure 3 molecules-29-05964-f003:**
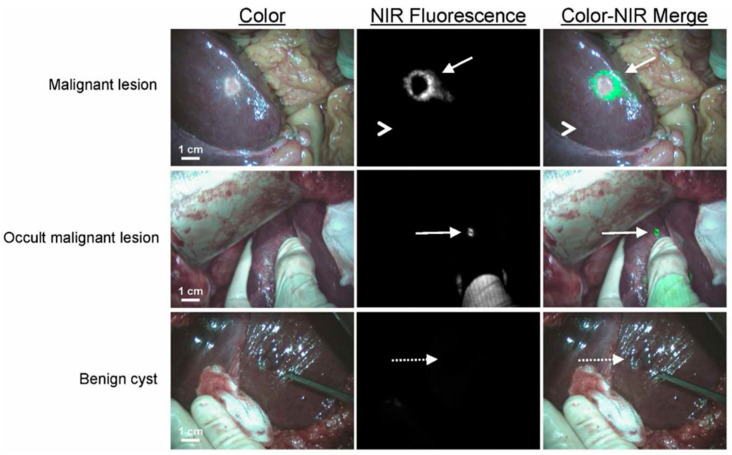
NIR fluorescence imaging of colorectal liver metastases. A colorectal liver metastasis (arrow) is clearly identified by a rim around the tumor in vivo (top row) 24 h after injection of 10 mg ICG. Normal liver tissue (arrowhead) shows minimal background uptake of ICG. In five patients, small, superficial, otherwise occult metastases (middle row, arrow) were identified by NIR fluorescence imaging. Benign lesions (bottom row, dashed arrow) could be differentiated from malignant lesions by a lack of a fluorescent rim around the lesion Reprinted with permission from Ref. [24], 2014, John Wiley & Sons Inc.

**Figure 4 molecules-29-05964-f004:**
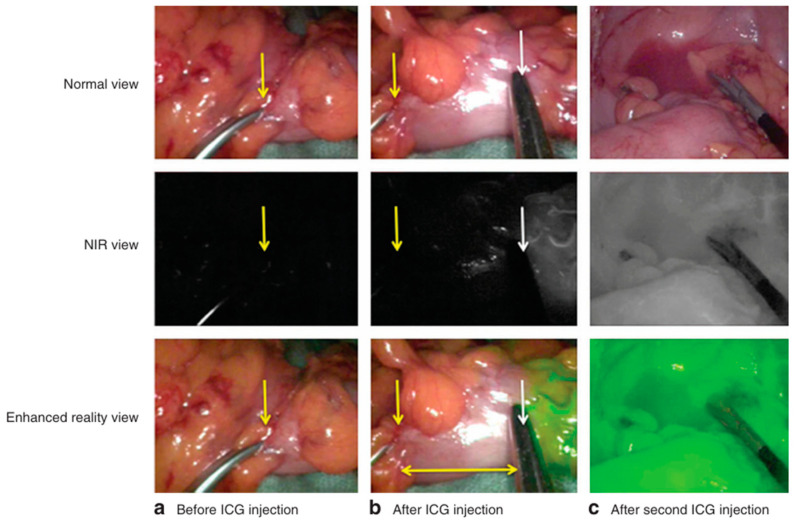
Near-infrared (NIR) perfusion assessment with change in plan owing to a lack of perfusion at the level of the screen originally planned. Intraoperative images are shown in normal view, NIR view, and enhanced reality view. (**a**) Image before indocyanine green (ICG) injection showing the planned area for proximal transection (yellow arrow) in a segment of descending colon after its mobilization (including high vascular ligation) and mesocolic preparation. (**b**) After ICG injection, a clear demarcation line appeared (white arrow) that was 4cm more proximal (vertical yellow arrow on the initial transection area) and led to more proximal transection (horizontal arrow shows distance that has been assessed as well perfused) being undertaken. (**c**) A second injection of ICG in the same patient showed satisfactory perfusion of the constructed colorectal anastomosis in situ [31]. Reprinted with permission from Ref. [31]. 2018, Oxford University Press.

**Figure 5 molecules-29-05964-f005:**
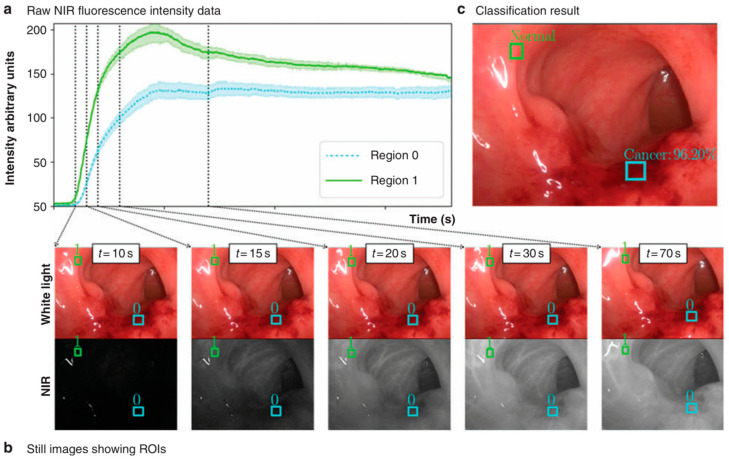
Representative image of process for intraoperative AI tissue classification illustrating cancer and normal regions of interest. (**a**) Plot of continuous measured raw near-infrared (NIR) fluorescence intensity data for 180s following administration of indocyanine green for two regions of interest (ROIs): blue trace (region 0) and green trace (region 1). (**b**) Still images from the white-light and fluorescence video with boxes showing ROIs for which data are being collected (top: white-light video sequence used for tissue tracking; bottom: corresponding NIR video sequence used for fluorescence intensity data acquisition). (**c**) Artificial intelligence classification results showing ROIs correctly classified as normal (green) and cancer (light blue), with classified regions superimposed on white-light video image [35] (Reprinted with permission from Ref. [35]. 2021, Oxford University Press.

**Figure 6 molecules-29-05964-f006:**
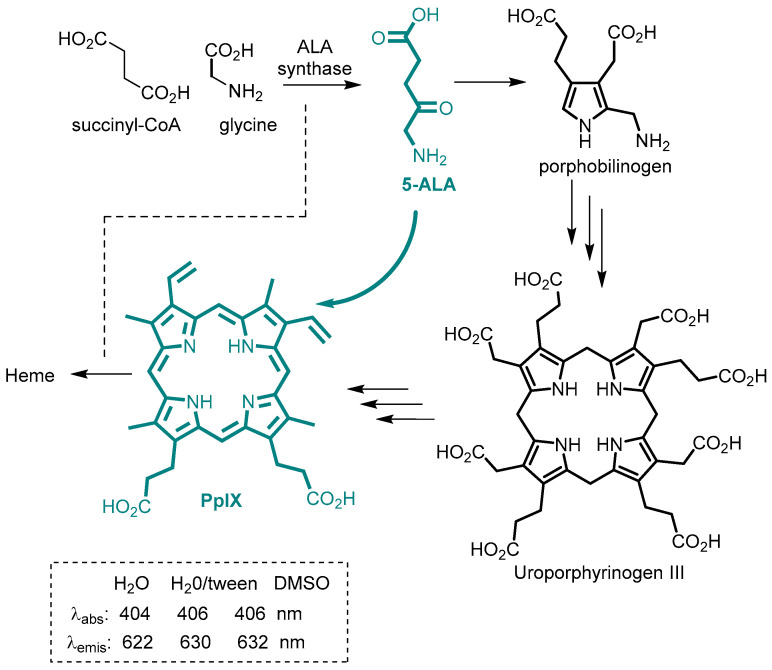
Partial heme biosynthetic pathway that produces protoporphyrin IX (PpIX) with negative feedback control indicated by the dotted line. Exogeneous introduction of 5-ALA leads to over production of PpIX (blue structures) via this pathway.

**Figure 7 molecules-29-05964-f007:**
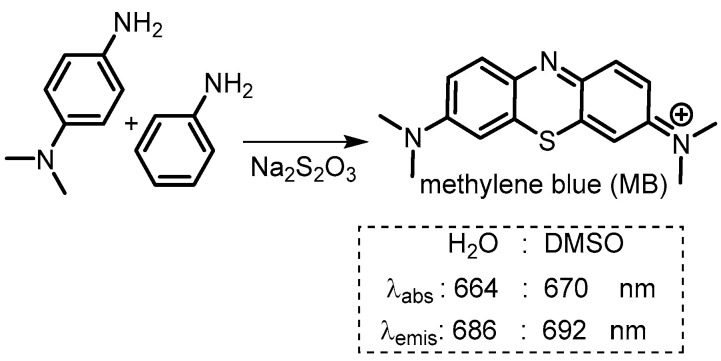
Synthesis and photophysical properties of methylene blue.

**Figure 8 molecules-29-05964-f008:**
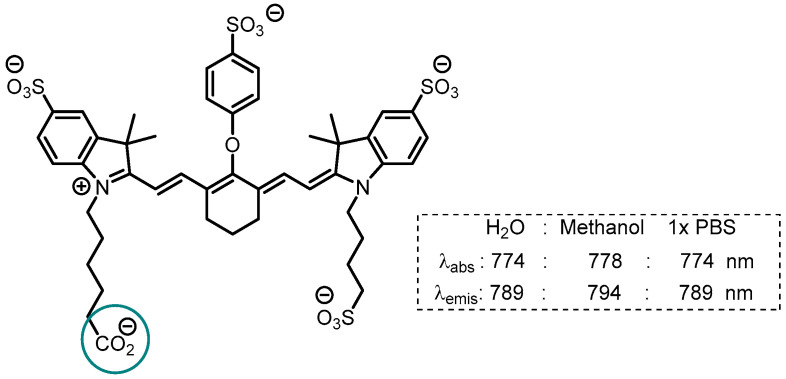
Structure of IR-Dye 800CW with position of conjugation indicated by blue circle, absorption and emission maxima values.

**Figure 9 molecules-29-05964-f009:**
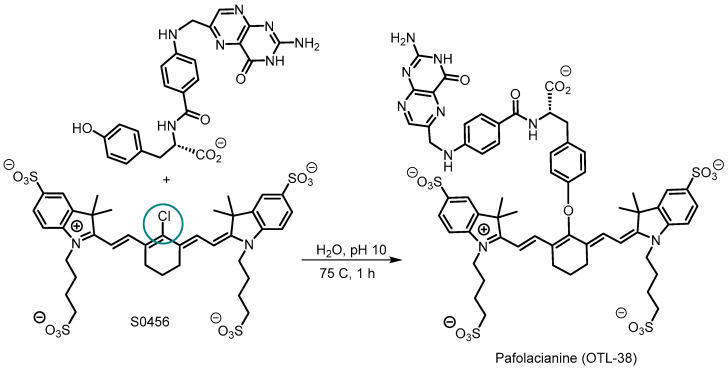
Synthesis of pafolacianine (OTL38) with position of conjugation to folic acid derivative indicated by blue circle.

## Data Availability

No new data were created.

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
