# Peer review of "Review of Clinically Assessed Molecular Fluorophores for Intraoperative Image Guided Surgery"

_molecules, 2024, doi:10.3390/molecules29245964_

Round 1
Reviewer 1 Report
Comments and Suggestions for Authors
This comprehensive review focuses on the application of molecular near-infrared (NIR) fluorophores for intraoperative image-guided surgery. It discusses their mechanisms, clinical applications, and the chemical optimization needed to improve performance. The study also addresses the integration of fluorescence-guided surgery (FGS) technology into standard surgical care, highlighting its current challenges and future prospects.
- The manuscript effectively addresses a growing interest in image-guided surgical innovations, aligning with current clinical needs.
- It covers multiple fluorophores, including FDA-approved agents like indocyanine green (ICG) and methylene blue (MB), and newer agents like pafolacianine.
- The discussion bridges technical chemical innovations with their practical clinical applications.
Author Response
Comments 1: This comprehensive review focuses on the application of molecular near-infrared (NIR) fluorophores for intraoperative image-guided surgery. It discusses their mechanisms, clinical applications, and the chemical optimization needed to improve performance. The study also addresses the integration of fluorescence-guided surgery (FGS) technology into standard surgical care, highlighting its current challenges and future prospects.
- The manuscript effectively addresses a growing interest in image-guided surgical innovations, aligning with current clinical needs.
- It covers multiple fluorophores, including FDA-approved agents like indocyanine green (ICG) and methylene blue (MB), and newer agents like pafolacianine.
- The discussion bridges technical chemical innovations with their practical clinical applications.
Response 1: Thank you for your comments. We appreciate your recognition of the comprehensive nature of our review and the significance of this topic. No specific changes are required in response to this comment.
Reviewer 2 Report
Comments and Suggestions for Authors
Dear Editor,
Thank you for the opportunity you have given me to review this manuscript - "Clinically assessed molecular near infrared fluorophores for intraoperative image guided surgery". I also congratulate the authors for the chosen topic, which is a newsworthy one.
However, certain changes need to be made in order to increase the value of the manuscript.
Thus, in the introduction section, paragraphs 37-70, it is necessary to add bibliographic indexes. The practical impact of what the authors propose to achieve through this review should be more clearly stated.
At the same time, the type of review should be specified from the title.
The bibliography is outdated and should be changed with references from the last 5 years.
Comments on the Quality of English LanguageEnglish language could be improved.
Author Response
Comments 1: Thank you for the opportunity you have given me to review this manuscript - "Clinically assessed molecular near infrared fluorophores for intraoperative image guided surgery". I also congratulate the authors for the chosen topic, which is a newsworthy one.
Response 1: Thank you for recognizing the importance of this topic.
Comments 2: However, certain changes need to be made in order to increase the value of the manuscript.
Thus, in the introduction section, paragraphs 37-70, it is necessary to add bibliographic indexes.
Response 2: Thank you for pointing this out. Additional bibliographic indexes have been added into the text as references [3], [4,5].
Comments 3: The practical impact of what the authors propose to achieve through this review should be more clearly stated.
Response 3: Additional text outlining the practical impact which we would like to achieve through this review has been added in Line 87-89.
Comments 4: At the same time, the type of review should be specified from the title.
Response 4: The title has been revised to explicitly indicate the type of review.
Comments 5: The bibliography is outdated and should be changed with references from the last 5 years.
Response 5: Additional references from the past 5 years have now been included in both the main text and the supporting information. While we have included a substantial number of recent references from the last 5 years to reflect current advancements, some older references have been retained as they represent significant milestones in the development of these techniques. These foundational works are crucial for providing context and understanding the evolution of the field, which is why we believe they remain important to include in the manuscript.
Comments 6: English language could be improved.
Response 6: Both authors have carefully reviewed the manuscript and made every effort to improve the quality of the English language.
Reviewer 3 Report
Comments and Suggestions for Authors
Abstract Evaluation Completeness: The abstract provides a concise overview of the paper's focus on near-infrared fluorescence guided surgery, highlighting its recent advancements and applications. It mentions the surgical goals, optimization of imaging agents, and the integration of this technology into routine care. However, it could benefit from a brief mention of specific fluorophores discussed in the paper to give readers a clearer idea of the content.
Background and Objectives: The introduction effectively sets the stage by discussing the historical context of fluorescence in medicine and the evolution of NIR fluorophores. It outlines the limitations of earlier fluorophores and emphasizes the advantages of NIR fluorescence, such as enhanced tissue penetration and reduced auto-fluorescence. The objectives of the review are clearly stated, focusing on the clinical trials and optimization of NIR fluorophores for surgical use.
Literature Search Methodology: The paper does not explicitly detail the literature search methodology, which is a critical aspect of a systematic review. While it discusses various NIR fluorophores and their applications, a more transparent description of how the literature was selected and analyzed would strengthen the review's credibility.
Discussion and References to Previous Articles: The discussion section effectively references previous studies and clinical trials involving NIR fluorophores. It highlights the advantages and limitations of these agents, providing a comprehensive overview of the current state of research. To enhance the discussion section and strengthen the scientific relevance of the article, I recommend integrating the following references: Cite the article with DOI: 10.3390/brainsci11060795 when discussing Gliolan and referencing Stummer’s work. This study highlights how the efficacy of Gliolan can vary not only based on histological characteristics but also due to molecular features.Include the review with DOI: 10.3390/cancers15164130 when addressing intraoperative fluorophores. This recent review provides a comprehensive discussion of various fluorophores used for intraoperative fluorescence, including ICG and experimental agents such as BLZ-100. It also highlights ongoing clinical trials.Incorporating these references would significantly enrich the discussion, connecting it to the latest research while broadening the clinical and molecular perspectives of the topics addressed
Conclusions: The conclusions drawn in the paper are consistent with the information presented throughout. They summarize the potential of NIR fluorophores in enhancing surgical outcomes and emphasize the need for further research to address existing challenges. The conclusions are sufficiently synthetic, encapsulating the key points discussed in the review while suggesting future directions for research.
Author Response
Comments 1: Abstract Evaluation Completeness: The abstract provides a concise overview of the paper's focus on near-infrared fluorescence guided surgery, highlighting its recent advancements and applications. It mentions the surgical goals, optimization of imaging agents, and the integration of this technology into routine care. However, it could benefit from a brief mention of specific fluorophores discussed in the paper to give readers a clearer idea of the content.
Response 1: Thank you for pointing this out. The fluorophores names have been added in the abstract.
Comments 2: Background and Objectives: The introduction effectively sets the stage by discussing the historical context of fluorescence in medicine and the evolution of NIR fluorophores. It outlines the limitations of earlier fluorophores and emphasizes the advantages of NIR fluorescence, such as enhanced tissue penetration and reduced auto-fluorescence. The objectives of the review are clearly stated, focusing on the clinical trials and optimization of NIR fluorophores for surgical use.
Response 2: Thank you for your thoughtful comments and for recognizing the effort we have put into organizing the paper’s framework and structure.
Comments 3: Literature Search Methodology: The paper does not explicitly detail the literature search methodology, which is a critical aspect of a systematic review. While it discusses various NIR fluorophores and their applications, a more transparent description of how the literature was selected and analyzed would strengthen the review's credibility.
Response 3: Thank you for your valuable feedback. In introduction section (Line 90-100), a comprehensive description of the search strategy used had now been included.
Comments 4: Discussion and  References to Previous Articles: The discussion section effectively references previous studies and clinical trials involving NIR fluorophores. It highlights the advantages and limitations of these agents, providing a comprehensive overview of the current state of research. To enhance the discussion section and strengthen the scientific relevance of the article, I recommend integrating the following references: Cite the article with DOI: 10.3390/brainsci11060795 when discussing Gliolan and referencing Stummer’s work. This study highlights how the efficacy of Gliolan can vary not only based on histological characteristics but also due to molecular features.Include the review with DOI: 10.3390/cancers15164130 when addressing intraoperative fluorophores. This recent review provides a comprehensive discussion of various fluorophores used for intraoperative fluorescence, including ICG and experimental agents such as BLZ-100. It also highlights ongoing clinical trials.Incorporating these references would significantly enrich the discussion, connecting it to the latest research while broadening the clinical and molecular perspectives of the topics addressed
Response 4: Thank you for your insightful suggestions and for recognizing the strengths of our discussion section. We appreciate the recommendation of those references, which we have carefully read them and found highly valuable for broadening our perspective. Accordingly, we have cited those two articles as reference 56 and 57.
Comments 5: Conclusions: The conclusions drawn in the paper are consistent with the information presented throughout. They summarize the potential of NIR fluorophores in enhancing surgical outcomes and emphasize the need for further research to address existing challenges. The conclusions are sufficiently synthetic, encapsulating the key points discussed in the review while suggesting future directions for research.
Response 5: Thank you for your positive feedback on the conclusions section.
Reviewer 4 Report
Comments and Suggestions for Authors
The paper is a comprehensive review of near-infrared (NIR) fluorescence-guided surgery, emphasizing its clinical applications, challenges, and advancements in imaging technologies. The authors focus on molecular NIR fluorophores such as Indocyanine Green (ICG), 5-Aminolevulinic Acid (5-ALA), Methylene Blue (MB), IR-Dye 800CW, and Pafolacianine. They explore these agents’ utility in intraoperative imaging for tumor identification, sentinel lymph node mapping, bile duct visualization, and tissue perfusion assessment. Challenges like dye specificity, tissue penetration, and false positives are highlighted, alongside the potential of artificial intelligence to enhance surgical precision. The review concludes with a call for further research to integrate fluorophores and imaging technologies into routine surgical workflows for improved patient outcomes.
The title is precise and reflects the content, but it could be rephrased to highlight the technological advancements in NIR-guided surgery more explicitly.
The abstract is well-structured but could benefit from specific quantitative data or notable achievements in clinical trials to provide a stronger impact.
The historical context of fluorescence is well-presented, but it consumes significant space that could be better allocated to discussing the relevance of fluorescence-guided surgery today.
The introduction could benefit from a clearer delineation of the review’s scope and objectives.
The discussion of individual fluorophores is detailed and supported by clinical data, but the paper lacks a clear comparative analysis of their strengths and limitations.
The review mentions FDA approvals, but it could benefit from discussing regulatory challenges and the timeline for approval processes for new fluorophores.
The section on artificial intelligence integration is promising but lacks depth in explaining how dynamic tissue perfusion analysis translates into clinical decisions.
The review should expand on the technological limitations, such as hardware compatibility across institutions.
While clinical trial results are extensively cited, the presentation feels fragmented. A summarized table comparing outcomes across fluorophores and applications could enhance readability.
There is little discussion on cost-effectiveness and accessibility of these technologies, which are critical for global adoption.
The challenges of false positives, tissue specificity, and administration delays are discussed but lack concrete recommendations for overcoming these barriers.
The review briefly mentions interdisciplinary collaboration (e.g., chemistry and medicine) but should explore this aspect more thoroughly to encourage innovative approaches.
Figures are informative but need consistent labeling and integration with the text. Some captions lack sufficient explanation for non-specialist readers.
Supplementary materials are referenced but not adequately summarized in the main text.
The review is descriptive and lacks a systematic methodology for selecting and analyzing studies.
The paper includes extensive citations, but some key references on recent AI applications in fluorescence-guided surgery are missing.
The writing is clear but occasionally verbose, especially in historical and descriptive sections.
There are minor grammatical errors and inconsistencies in formatting, particularly in figure captions and references.
The conflict of interest statement is transparent, but it should elaborate on the measures taken to ensure objectivity in the review, given the financial interests of one author.
Author Response
Comments 1: The paper is a comprehensive review of near-infrared (NIR) fluorescence-guided surgery, emphasizing its clinical applications, challenges, and advancements in imaging technologies. The authors focus on molecular NIR fluorophores such as Indocyanine Green (ICG), 5-Aminolevulinic Acid (5-ALA), Methylene Blue (MB), IR-Dye 800CW, and Pafolacianine. They explore these agents’ utility in intraoperative imaging for tumor identification, sentinel lymph node mapping, bile duct visualization, and tissue perfusion assessment. Challenges like dye specificity, tissue penetration, and false positives are highlighted, alongside the potential of artificial intelligence to enhance surgical precision. The review concludes with a call for further research to integrate fluorophores and imaging technologies into routine surgical workflows for improved patient outcomes.
Response 1: Thank you for your thorough summary and feedback.
Comments 2: The title is precise and reflects the content, but it could be rephrased to highlight the technological advancements in NIR-guided surgery more explicitly.
Response 2: Thank you for your suggestion. We agree on the importance of highlighting the technological advancements of NIR-guided surgery. Given the word limitation of the title, we have revised and emphasized these advancements more explicitly in the abstract to ensure they are adequately addressed.
Comments 3: The abstract is well-structured but could benefit from specific quantitative data or notable achievements in clinical trials to provide a stronger impact.
Response 3: Thank you for your feedback. Additional text has been added to the abstract to call out past and future notable achievement of fluorescence guided surgery.
Comments 4: The historical context of fluorescence is well-presented, but it consumes significant space that could be better allocated to discussing the relevance of fluorescence-guided surgery today.
Response 4: We appreciate your observation regarding the historical context. Our intention was to provide a foundation for understanding the evolution of each fluorophore for the readership’s benefit. We also recognize the importance of focusing on the current relevance of fluorescence-guided surgery. Based on your suggestion, we have revisited the historical section to make it more concise, reallocating space to emphasize the advancements of this technology.
Comments 5: The introduction could benefit from a clearer delineation of the review’s scope and objectives.
Response 5: Thanks for highlighting this. The introduction section has been revised to better delineate its scope and objectives.
Comments 6: The discussion of individual fluorophores is detailed and supported by clinical data, but the paper lacks a clear comparative analysis of their strengths and limitations.
Response 6: Thank you for pointing this out. We agree that a comparative analysis of the strengths and limitations of the fluorophores would benefit the readership, however, since different fluorophores are employed in distinct clinical scenarios, with varying complexities and some aspects still not fully understood, a direct comparison may not provide an accurate or meaningful assessment. Instead, we prefer to focus on discussing each fluorophore within its specific context to provide a clearer understanding of its unique applications and challenges.
Comments 7: The review mentions FDA approvals, but it could benefit from discussing regulatory challenges and the timeline for approval processes for new fluorophores.
Response 7: Thank you for your suggestion. This part has been added in the conclusion (Line 582-584) section of the manuscript.
Comments 8: The section on artificial intelligence integration is promising but lacks depth in explaining how dynamic tissue perfusion analysis translates into clinical decisions.
Response 8: Thank you for your recognition of the potential of utilizing artificial intelligence in FGS. We have added in more detail to explain how dynamic tissue perfusion analysis translates into clinical decisions in the main text of the manuscript at Line 350-356.
Comments 9: The review should expand on the technological limitations, such as hardware compatibility across institutions.
Response 9: Thank you for your suggestion. Some discussion about hardware compatibility has been added in conclusion at Line 497-504.
Comments 10: While clinical trial results are extensively cited, the presentation feels fragmented. A summarized table comparing outcomes across fluorophores and applications could enhance readability.
Response 10: Thank you for your valuable suggestion. Some details regarding the applications or conditions are summarized in the supplementary materials of Table 1-8.
Comments 11: There is little discussion on cost-effectiveness and accessibility of these technologies, which are critical for global adoption.
Response 11: Thanks for your suggestion. Additional text has been added discussing cost-effectiveness and accessibility in the conclusion section at Line 560-565.
Comments 12: The challenges of false positives, tissue specificity, and administration delays are discussed but lack concrete recommendations for overcoming these barriers.
Response 12: We agree that overcoming challenges such as false positives, tissue specificity or administration delays is a crucial topic. As such, we hope this review will encourage more interdisciplinary collaboration among researchers to address these barriers and develop solutions.
Comments 13: The review briefly mentions interdisciplinary collaboration (e.g., chemistry and medicine) but should explore this aspect more thoroughly to encourage innovative approaches.
Response 13: Thanks for your suggestion. More content about interdisciplinary collaboration has been added in the conclusion section at Line 579-582.
Comments 14: Figures are informative but need consistent labeling and integration with the text. Some captions lack sufficient explanation for non-specialist readers.
Response 14: Thanks for your feedback. We have carefully reviewed the entire article to ensure consistent labeling across all figures and to improve their integration with the text. Since some figures are cited from other references, we are unable to modify their captions. However, we have provided additional explanation in text to ensure a clearer understanding for non-specialist readers.
Comments 15: Supplementary materials are referenced but not adequately summarized in the main text.
Response 15: Thanks for pointing this out. We have revised to include a more comprehensive summary of the supplementary materials in the part of “Supplementary Materials”.
Comments 16: The review is descriptive and lacks a systematic methodology for selecting and analyzing studies.
Response 16: Thank you for your valuable feedback. In introduction section, we have added a more comprehensive description of the search strategy at Line 90-100.
Comments 17: The paper includes extensive citations, but some key references on recent AI applications in fluorescence-guided surgery are missing.
Response 17: Thank you for pointing this out. Another reference (Reference 33) has been added in regarding AI based analysis method of ICG angiography at Line 323-327.
Comments 18: The writing is clear but occasionally verbose, especially in historical and descriptive sections.
Response 18: Thank you for your feedback. We have reviewed the whole article to make it more concise, especially the historical and descriptive section.
Comments 19: There are minor grammatical errors and inconsistencies in formatting, particularly in figure captions and references.
Response 19: Thank you for your feedback. We have carefully reviewed the manuscript again to correct any grammatical errors and ensure consistency in formatting.
Comments 20: The conflict of interest statement is transparent, but it should elaborate on the measures taken to ensure objectivity in the review, given the financial interests of one author.
Response 20: Thank you for your feedback. The section of “Conflicts of Interest” has been revised to include additional details on the specifics of the conflict of interest. Also, a detail description of the literature search strategy used had now been included in the introductions section at Line 90-100.
Round 2
Reviewer 3 Report
Comments and Suggestions for Authors
the authors have addressed the points
Reviewer 4 Report
Comments and Suggestions for Authors
After a through revision according to the reviewer's suggestion, the paper has been notably improved and now it is ready for publication. Thank you.